



# Large, old pools of carbon and microbial communities are present deep in soils under a temperate planted forest

Alexa K. Byers[1], Loretta G. Garrett[2], Charlotte Armstrong[3], Fiona Dean[2], Steve A. Wakelin[3]

[1]Bioprotection Aotearoa, PO Box 85084, Lincoln University, Lincoln, 8014, New Zealand
[2]Scion, Private Bag 3020, Rotorua, 3046, New Zealand
[3]Scion, PO Box 29237, Riccarton, Christchurch, 8440, New Zealand

*Correspondence to*: Alexa K. Byers (alexa.byers@lincoln.ac.nz)

**Abstract.** Forest soils are fundamental in regulating the global carbon (C) cycle; their capacity to accumulate large stores of C means they are vital in mitigating the effects of climate change. Understanding the processes that regulate forest soil organic C (SOC) dynamics and stabilisation is important to maximise the capacity and longevity of C sequestration. Compared to surface soil layers, little is known about the SOC dynamics in subsoil layers, *sensu* those below 30 cm depth. This knowledge
gap creates large uncertainties when estimating the global distribution and vulnerability of SOC reserves to climate change. This study aimed to dive deep into the subsoils of Puruki Experimental Forest (New Zealand) and characterise the incremental changes in SOC dynamics and the soil microbiome down to 1 metre soil depth. ITS and 16S rRNA sequencing and quantitative real-time PCR were used to measure changes in soil microbial diversity, composition, and abundance. Stable ($\delta^{13}$C) and radioactive ($^{14}$C) C analyses were performed to assess depth-driven changes in SOC stability and age. We conservatively
estimate more than 35% of total C stocks are present in subsoil layers below 30 cm. Although C age steadily increased with depth, reaching a mean radiocarbon age of 1571 yBP (years before present) in the deepest soil layers, the stability of SOC varied between different subsoil depth increments. Declines in soil carbon were associated with lower microbial diversity, abundance, and significant shifts in community membership. These research findings highlight the importance of quantifying subsoil C stocks for accurate systems-level global and local C budgets and modeling. Furthermore, performing a broad range
of analytical measures (i.e. $^{13}$C & $^{14}$C natural abundance, and microbiome analysis) is vital to assess the vulnerability of subsoil C to climate change.

## 1 Introduction

Soils can store huge amounts of carbon (C), with the global soil C pool of 2500 gigatons (Gt) nearly double that of the atmospheric and biotic pools combined (Lal, 2004). Forest soils are influential in regulating the global C cycle and have been
estimated to function as a net C sink of $-7.6 \pm 49$ GtCO$_2$e yr$^{-1}$ (Harris et al., 2021). Forest soils account for over 70% of the world's soil organic C (SOC) reserves, and support over 80% of the total aboveground terrestrial C stocks (Batjes, 1996; Jobbágy and Jackson, 2000; Six et al., 2002). Developing forest management strategies that maximise soil C sequestration





will, therefore, serve as a powerful tool for offsetting fossil fuel emissions and mitigating against climate change (Jandl et al., 2007; Mukul et al., 2020).

Compared with surface soil layers, deeper subsoil layers of forest ecosystems are underexplored environments (Yost and Hartemink, 2020). Although the absolute concentration of soil organic C (SOC) declines with depth (Schmidt et al., 2011), extensive deposits of previously unquantified SOC are often present in deeper soil layers (Gonzalez et al., 2018; Ross et al., 2020). Indeed, it has been reported that approximately 50% of total SOC may be contained within soil layers below 30 cm (Balesdent et al., 2018; Jobbágy and Jackson, 2000).

SOC in deep soil is typically ancient, having a mean global age calculated at over 8000 years (Shi et al., 2020). When undisturbed, deep SOC stocks are considered to represent the 'stable' fraction of the SOC pool, having a physical and chemical nature that is, while *in situ* conditions remain constant, resistant to decomposition by soil microorganisms (Schmidt et al., 2011). However, the stability of sequestered SOC is complex and can be related to a range of variables including soil clay content, mineralogy, structure and texture, landscape position, association with micro aggregates, climate, and vegetation

(Jobbágy and Jackson, 2000; Lal, 2004). Furthermore, the stability of SOC is strongly influenced by the activities of plant roots and, thereby, soil microorganisms (Kuzyakov, 2010; Song et al., 2020). The historical biases against sampling deeper soil layers mean we have a limited understanding of how the properties of subsoil environments influence their ability to sequester and store SOC. This creates huge uncertainty when modeling the stability of deep SOC pools to climate change (Gross and Harrison, 2019).

Soil microorganisms have a key role in the transformation and stabilisation of SOC (Lladó et al., 2017; Spohn et al., 2016) and are therefore influential in determining patterns of C sequestration (Jastrow et al., 2007). Studying the fundamental changes that occur to soil microbial communities with depth is essential to understand the extent and nature of their role in regulating long-term SOC dynamics. Previous studies have reported that soil microbial diversity, abundance, respiration, and C turnover decline with depth (Blume et al., 2002; Eilers et al., 2012; Fang et al., 2005; Fierer et al., 2003; Spohn et al., 2016), and more

recent research efforts have explored the taxonomic and functional changes that occur in microbial communities with soil depth (Brewer et al., 2019; Frey et al., 2021; Hao et al., 2021). However, our knowledge of the response of soil microbial communities to depth and the environmental properties driving these changes is still poorly understood, particularly for coniferous forest ecosystems. Furthermore, the inherently heterogeneous nature of soil environments, at both global and local scales (Curd et al., 2018; Kuzyakov and Blagodatskaya, 2015; Štursová et al., 2016), means context-specific research is

required to understand the SOC dynamics and associated microbiota of forest subsoil environments. To maximise the sink capacity of forest soils, and perhaps even to retain the SOC already stored at depth, we need to better understand the processes that regulate C sequestration and long-term C storage within the deep soil environment.

Using Puruki Experimental Forest (central North Island, New Zealand) as an exemplar case study (Garrett et al., 2021), this research aimed to improve our fundamental understanding of subsoil C dynamics within a production forest. Using a wide

range of different analytical methods and procedures, we aimed to examine the variability in soil C dynamics at a highly refined spatial scale. Changes in the diversity, composition, and abundance of bacterial and fungal communities with depth were




quantified using 16S rRNA/ITS amplicon sequencing and quantitative real-time PCR (qPCR). Measurements of natural $^{13}$C abundances in soils have previously been used to study SOC dynamics, with the isotopic composition ($^{13}$C/$^{12}$C ratio) of soils functioning as a proxy for soil C turnover (Garten et al., 2008; Paul et al., 2020; Poage and Feng, 2004; Wang et al., 2018).

Therefore, to better understand changes in C cycling dynamics down each soil core, the isotopic composition of each depth increment was measured. Radiocarbon aging using $^{14}$C abundance was further used to aid the identification of older stabilised C stocks (Chabbi et al., 2009).

## 2 Materials and Methods

### 2.1 Study site and soil collection

Soil for this study was sampled from Puruki Experimental Forest (38° 26'S, 176° 13'E), located in the Paeroa Range of New Zealand's central North Island (Figure A1). Puruki Experimental Forest (henceforth referred to as Puruki Forest), is a 35-hectare *Pinus radiata* forest. Puruki Forest was established in 1973 after previously being converted from a podocarp/hardwood native forest to ryegrass and clover pasture in 1957 (Beets and Brownlie, 1987; Garrett et al., 2021). Puruki Forest is now near the end of its second rotation, having previously been harvested in 1997 (Oliver et al., 2004). Being

extensively previously studied, Puruki Forest was selected for this study as an extensive body of historical data available related to soil health, productivity, and land management is available; these factors are important for helping us understand drivers of subsoil C dynamics (Garret et al., 2021).

Puruki Forest sits at an elevation range of 500 to 700m a.s.l., varies from gently (less than 12°) to steeply sloping (up to 30°) land, with a mean annual rainfall of 1500 mm and mean annual temperature of 10°C (Beets and Brownlie, 1987; Brownlie and

Kelliher, 1989). These soils are highly permeable, composed of loamy sand, silty sand, and gravel (Taupo lapilli). Soil parent materials originated from Taupo volcanic centre (1850 ± 100 BP) and older ash showers from Taupo and Okataina volcanic centres and are classified as Orthic Pumice soils (New Zealand Soil Classification System; (Hewitt, 2010; Rijkse and Bell, 1974).

A permanent sample plot (PSP) within the Rua sub-catchment of Puruki Forest was targeted for sampling. Soils were collected

immediately adjacent to the plot to avoid disturbance of ongoing longer-term monitoring within the PSP itself. For sampling, 10 points spaced 2 metres (m) apart were set out along an 18 m transect. At each sampling point, two paired 1 m deep soil cores were extracted using a motorised percussion soil sampler capable of collecting intact cores with an internal plastic sleeve; one core was taken for bulk density analysis, and the other was used for DNA and chemical analysis. Following extraction, core pairs were visually compared to check for any considerable differences in soil colour and texture by depth (Figure A2).

Intact soil cores were transported back to the lab in cool, dark conditions. Care was taken to minimise disruption during transport (cores all remained intact from field to lab). Following transport, soil cores were divided into 10 cm increment samples. Incidences of compaction which occurred during sample collection were adjusted for during the division of cores into depth increments (Supplementary Methods).



## 2.2 Sample preparation

To prepare soils for DNA analysis, moist soil samples were sieved to less than 2 mm and stored at 4°C before DNA extraction. Soil DNA was extracted from 0.25 g of each soil sample using a DNeasy Powersoil Pro Kit (Qiagen) according to the manufacturer's instructions. Soils required for measurements of total C & nitrogen (N), total phosphorus (P), Bray P, pH, Mehlich 3 extractable elements, cation exchange capacity (CEC), exchangeable cations, organic P, and inorganic P analysis were air dried and sieved to <2 mm. For measurements of bulk density, air-dried fine (<2 mm) and coarse (>2 mm) soil

fractions were oven dried separately at 105°C to a constant weight. Subsamples of air-dried soils were finely ground using a Spex800D ball mill for measurements of total C (%), total N (%), $\delta^{13}C$ [‰], and radiocarbon $^{14}C$ analysis [‰].

## 2.3 Soil chemistry

Total C and N were determined on fine (<2 mm) and coarse (>2 mm) fraction fine ground soils by dry combustion using a LECO CNS-2000 carbon nitrogen analyser (Rayment and Lyons, 2011). Soil pH was determined in water at a soil: water ratio

of 1:2.5 (Blakemore et al., 1977). Total phosphorus (P) was measured by flow injection analyser (FIA) colorimetry after sulphuric acid digestion (Blakemore et al., 1977; Taylor, 2000). Bray P was measured by FIA colorimetry after sequential Bray 2 ($NH_4F/HCl$) extraction at a ratio of 1:10 soil: extractant (Blakemore et al., 1977; Bray and Kurtz, 1945). Exchangeable cations (Ca, K, Mg, and Na) were measured by inductively coupled plasma mass spectrometry (ICP-MS) after 1:50 (macro) $NH_4CH_3COO$ leaching. Cation exchange capacity (CEC) was measured by FIA colorimetry after 1M $NH_4COOH$ leaching,

followed by 1M NaCl (Blakemore et al., 1977; Sparks et al., 1996).

### 2.3.1 Stable isotope ($\delta^{13}C$) analysis

Fine ground soil samples were sent to GNS Stable Isotope Laboratory (Wellington, New Zealand) for $\delta^{13}C$ analysis by isotope ratio mass spectrometry. Soil samples were pre-treated by adding 10% HCl to 1 g of each sample and left overnight at room temperature. Samples were then centrifuged and rinsed to neutral pH. Following pre-treatment, samples were freeze-dried and

measured on an Isoprime mass spectrometer using a Eurovector elemental analyser. All $\delta^{13}C$ results are reported with respect to VPDB and N-Air, normalized to GNS internal standards which were cane sugar (-10.3‰), beet sugar (-24.6‰), and EDTA (-31.1‰). The stable carbon isotope composition ($^{13}C/^{12}C$) is reported in $\delta^{13}C$ values presented as per thousand [per mille, ‰], with an analytical precision of 0.2‰.

### 2.3.2 Radiocarbon $^{14}C$ analysis

Fine ground soil samples were graphitized at the Houghton Carbon, Water and Soils Lab (USDA Forest Service Northern Research Station, Michigan, US), and radiocarbon measurements were conducted at the DirectAMS facility (Bothell, Washington, US). Soil samples were combusted at 900°C for 6 hours in evacuated quartz tubes with cupric oxide (CuO) and silver (Ag) wire to form $CO_2$ gas. After combustion, $CO_2$ was reduced to graphite through heating at 570°C with hydrogen



(H$_2$) gas and an iron (Fe) catalyst (Vogel et al., 1987). Graphite targets were analysed for radiocarbon abundance using
accelerator mass spectrometry (AMS) and corrected for mass-dependent fractionation using measured δ$^{13}$C values according
to Stuiver and Polach (1977).

## 2.4 Microbial amplicon sequencing

Following Earth Microbiome Project (EMP) protocols, barcoded forward and reverse primers 515F/806R were used to amplify
the 16S rRNA V4 region (Caporaso et al., 2012). Nested PCRs were performed on DNA samples that initially failed to amplify
using the non-barcoded primers 27F/1492R. Following nested PCR, amplicons were amplified using barcoded 515F/806R
primers. The EMP's standard ITS amplicon protocol was followed to amplify the fungal ITS gene region using the barcoded
primers ITS1f /ITS2 (Bokulich and Mills, 2013; Hoggard et al., 2018). DNA samples that initially failed to amplify were
targeted for nested PCR using primers NSA3/ITS4 and nested PCR products were PCR amplified using barcoded primers
ITS1f/ITS2. Full details of PCR reaction protocols are provided in the Supplementary Materials. PCR amplicons were sent for
Illumina 500 MiSeq 250bp (16S) and Illumina 600 MiSeq 300 bp (ITS) paired-end sequencing at the Australian Genome
Research Facility (AGRF; Melbourne, Australia). For the 16S amplicon libraries, eight no-template control (NTC) samples,
which showed no visible bands during agarose gel electrophoresis, were sent for sequencing to check for contaminants.

## 2.5 Quantitative PCR

The absolute concentration of bacterial and fungal DNA in each soil DNA sample was determined from standard curves using
AriaMx SYBR Green qPCR (Agilent Technologies). Broad range forward and reverse primers 338F/518R and ITS1F/5.8s
were selected to target the 16S rRNA (V3-V4) and ITS gene region for quantification (Shahsavari et al., 2016). Full details of
the qPCR reaction protocol and standard curve preparation are given in the Supplementary Methods. Following quantification,
16S and ITS rRNA copy numbers were adjusted to copy numbers per ng DNA.

## 2.6 Statistical analysis

Prior to analysis, slope corrected C stocks (Mg ha$^{-1}$) were calculated based on total C (%) and soil bulk density measurements.
The values for total C & N and C stocks are reported using the sum of <2 mm and >2 mm soil fractions. Values for Bray P
sequential 1 and exchangeable cations (K, Ca, Mg, Na) were slope corrected and converted to kg ha$^{-1}$. The changes in soil pH,
total C (Mg ha$^{-1}$), total N (kg ha$^{-1}$), C:N ratio, δ$^{13}$C [‰], Bray P (kg ha$^{-1}$), total bulk density (g cm$^{-3}$), and exchangeable cations
K, Ca, Mg, and Na (kg ha$^{-1}$) with soil depth were tested for significance using non-parametric Friedman's tests with soil core
transect position as the blocking variable. *Post hoc* tests were performed using pairwise Wilcoxon rank sum tests with
Bonferroni's correction.

To calculate differences in the allocation of C stocks down each soil profile, the proportion of C (Mg ha$^{-1}$) allocated to each
depth increment was calculated relative to the sum of total C (Mg ha$^{-1}$) for each sample core. Differences in 16S and ITS rRNA
copy number with depth were tested for significance using the Kruskal Wallis chi-squared test, with post hoc testing performed





using Dunn's test with Bonferroni's adjustment. To estimate the allocation patterns of microbial biomass down the soil profile, the proportion (%) of 16S and ITS copy numbers allocated to each depth increment was calculated relative to the total sum of the soil core.

### 2.6.1 Microbial diversity analysis

Base calling and demultiplexing of sequencing libraries to per-sample fastQ files was performed using Illumina's bcl2fastq
software (version 2.20.0.422). The DADA2 version 1.18 workflow (Callahan et al., 2016) was used to process paired-end fastQ files into amplicon sequence variants (ASVs). Briefly, forward and reverse reads were quality filtered, trimmed, and denoised before being merged into ASVs. Chimeric ASVs were removed, and taxonomies were assigned to ASVs using the Ribosomal Database Project (RDP) Classifier (Wang et al., 2007) and the UNITE (Abarenkov et al., 2021) databases. After DADA2 processing, $74.92 \pm 5.60\%$ of 16S input reads and $44.46 \pm 14.35\%$ of ITS input reads were retained. Following
DADA2 processing, ASV tables were filtered to remove unidentified phyla, unwanted phyla (i.e. Cyanobacteria/Chloroplasts), and singletons. For the 16S dataset, ASVs which had a higher average count in NTC samples than DNA samples were also removed. Rarefaction curves showing ASV richness by depth increment and PCR type can be seen in Figures A.3 and A.4. Following the removal of samples with low ASV counts, ASV tables were rarefied to even sampling depth before analysis (16S: 11613 ASVs per sample, ITS: 10101 ASVs per sample).
Microbial alpha and beta diversity analyses were calculated on rarefied ASV tables using the R packages phyloseq (Mcmurdie and Holmes, 2013) and vegan (Oksanen et al., 2020). Changes in alpha diversity with depth were tested for statistical significance using Kruskal Wallis chi-squared tests as described previously. Pairwise Spearman's correlation tests with Bonferroni's adjustment were used to correlate alpha diversity values with soil chemical parameters using psych R (Revelle, 2021). Bray-Curtis distance matrices were calculated to measure community dissimilarity and ordinated using non-metric
multidimensional scaling (NMDS). Differences in community dissimilarity by depth were tested for significance using PERMANOVA, and pairwise differences were tested using pairwiseAdonis R (Arbizu, 2017), again corrected for false discovery with Bonferroni's adjustment.

For both alpha and beta diversity analyses, the impacts of transect sampling position and nested vs single PCR type were tested for significance. Mantel tests were used to correlate Euclidean distance matrices of soil physicochemical properties to Bray-
Curtis matrices of microbial community composition. The envfit function (vegan R) was used to fit soil properties that significantly correlated to community dissimilarity onto NMDS ordination plots. Venn diagrams were produced to show the proportion (%) shared vs unique ASVs between depth increments using MicEco R (Russel, 2021).

Analysis of Compositions of Microbiomes with Bias Correction, ANCOM-BC (Lin and Peddada, 2020) was used to identify microbial taxa which had a significant log-fold change between topsoil (*a priori* defined as 0 to 30 cm) and subsoil layers
(below 30 cm). Hierarchical clustering was performed on Euclidean distances of log adjusted abundances, using the average linkage method (hclust R). Dendrograms were constructed using ggtree (Yu et al., 2017) to visualise the hierarchical clustering patterns of microbial orders based on their log-adjusted abundances. The log-adjusted abundances of microbial taxa obtained

from ANCOM-BC analysis were correlated to soil chemical properties using pairwise Spearman's correlation tests with Bonferroni adjustment using psych R (Revelle, 2021). To assess changes in the diversity of high-level 16S rRNA functional genes with soil depth, PICRUSt2 (Douglas et al., 2020) was used to predict MetaCyc pathway abundances (Caspi et al., 2018). The results of the MetaCyc pathway abundance predictions were analysed in phyloseq R using the same alpha and beta diversity methods as outlined above.

### 2.6.2 Microbial network analysis

ASV tables were split into samples from topsoil layers (as before) to obtain core communities specific to each soil layer. Following this, ASV tables were filtered to retain only those with a total relative abundance of at least 0.05% in a minimum of 3 samples and then summarised to genus level.

SPIEC-EASI networks were constructed using the Meinshausen–Buhlmann (MB) neighbourhood algorithm in SpiecEasi R (Kurtz et al., 2015). Negative edge weights were set to zero to retain only positive associations. Networks were visualised using igraph R (Csárdi and Nepusz, 2006) and network statistics were calculated using Cytoscape 3.8.2 (Shannon et al., 2003). Significant differences (p<0.05) in key network statistics between topsoil and subsoil communities were tested using Wilcoxon signed rank tests (Mundra et al., 2021). Descriptions of these network statistics can be found in the Supplementary Methods.

## 3 Results

### 3.1 Soil chemistry

Total C (Mg ha$^{-1}$), total N (kg ha$^{-1}$), C:N ratio, and exchangeable cations K, Ca, Mg, and Na (kg ha$^{-1}$) significantly declined with soil depth (p<0.05), whereas soil pH and bulk density (g cm$^{-3}$) significantly increased with soil depth (p<0.05).

### 3.2 Allocation of C stocks and microbial biomass

Total average SOC stocks (<2 mm and >2 mm fractions) across the soil cores were 99.71 ± 28.78 Mg ha$^{-1}$ C to 1 m soil depth. Only 1.88 ± 1.92 Mg ha$^{-1}$ of this C was present in the coarse (>2 mm) soil fraction and this did not vary by soil depth increment (ANOVA: F-value = 1.58; p>0.05). On average across the 10 soil cores, 34.58 ± 10.25% of the total C stocks were allocated to the 30 to 100 cm (subsoil) layer (Figure 1; Table A2). The amount of total C allocated to subsoil varied, ranging from 46.60% (transect position 0 m) to 16.21% (transect position 18 m). The allocation patterns of total C are associated strongly with the distribution of microbial biomass down the soil profile, with 33.45 ± 10.92% of fungal biomass and 49.11 ± 4.82% of bacterial biomass being allocated to subsoils (Figure 1; Tables A3 and A4).



### 3.3 $\delta^{13}$C and $^{14}$C natural abundance measurements

Overall, $\delta^{13}$C [‰] significantly increased by 1.58‰ (p<0.05; Table A1) between depth increments of 0 to 10 and 90 to 100 cm. However, this trend was variable. In particular, for samples in the depth layer from 70 to 90 cm, the enrichment of $^{13}$C was lower than the rest of the subsoil (Figure 3). Values for $^{14}$C [‰] and CRA [yBP] of SOC declined with depth (Figure 2; Table A1). The age of SOC steadily increased with depth and reached a mean age of 1570.83 ± 316.28 yBP in the deepest soil layers of 90 to 100 cm.

### 3.4 Changes in microbial diversity, abundance, and composition with depth

The richness, diversity, and evenness of fungal and bacterial communities were all highest in the top surface horizons and significantly declined with depth (p<0.05; Figure 3; Table A6 and A7). The same trends were evident for the predicted abundances of bacterial metabolic cycles (MetaCyc pathways ; p<0.05) with depth (Table A7). Transect position had no relationship to bacterial alpha diversity metrics, however, fungal evenness did vary among the sample cores (Table A7). In

samples that required nested PCR to achieve amplification of amplicons, microbial richness and diversity were lower than that of samples that required a single PCR (p< 0.05). It is unknown if this is related to the repeated use of PCR itself, or a reflection of the lower microbial abundance in these samples being associated with reduced species diversity.

The abundances of both bacteria and fungi significantly declined with soil depth (Figure 3; Table A7) but were unaffected by transect position. Alike with alpha diversity, most of the significant differences in 16S and ITS copy number occurred between

the upper 0 to 20 cm depth increment when compared with the rest of the soil; few significant pairwise differences were evident among soil layers sampled below 30 cm (Figure A7).

When tested using pairwise Spearman's rank correlations (Figure 4), microbial alpha diversity and abundance were strongly positively correlated to total C, total N, C:N ratio, and $^{14}$C (rho > 0.78; p<0.05). Soil pH was strongly negatively correlated to microbial alpha diversity and abundance (rho > - 0.89; p<0.05). $\delta^{13}$C was strongly negatively correlated to fungal richness

(rho= -0.84, p<0.05), and bulk density (g cm$^{-3}$) were strongly negatively correlated to bacterial abundance (rho= -0.90, p<0.05). Bacterial (PERMANOVA: $R^2$ = 0.21, p<0.001) and fungal ($R^2$ = 0.26, p<0.001) community composition varied significantly between depth increments (Figure 5). Pairwise tests identified that differences in community composition in soils sampled from 0 to 30 cm contributed to the majority of significant differences between groups; i.e. differences in microbial composition between soil increments below 30 cm were non-significant (Figure A9).

The composition of bacterial and fungal communities significantly differed by sampling transect position (bacteria: $R^2$ = 0.13, p<0.001; fungi: $R^2$ = 0.04, p<0.001), and PCR type (bacteria: $R^2$ = 0.14, p<0.001; fungi: $R^2$ = 0.06, p<0.001). However, these factors accounted for less variation in community composition than soil depth. In addition, the composition of 16S functional genes was significantly different between soil depth increments ($R^2$ =0.34, p<0.001; Figure A10). However, transect position ($R^2$ = 0.13, p<0.001) and PCR type ($R^2$ = 0.09, p<0.001) were also significant factors influencing differences in the composition

of bacterial functional genes.





Mantel tests identified that $^{14}$C (R= 0.38, p<0.001), C:N ratio (R= 0.18, p<0.05), bulk density (R= 0.20, p<0.05), $\delta^{13}$C (R= 0.13, p<0.05), total N (R= 0.13, p<0.05), and exchangeable Ca (R= 0.12, p<0.01) significantly correlated to differences in bacterial community composition (Figure 5). Total N (R= 0.17, p<0.01), exchangeable Ca (R= 0.13, p<0.01), bulk density (R= 0.12, p<0.05), $\delta^{13}$C (R= 0.18, p<0.01), and $^{14}$C (R= 0.38, p<0.001) were also significantly correlated to differences in the

composition of bacterial functional genes (Figure A10). In addition, $^{14}$C (R= 0.32, p<0.001), $\delta^{13}$C (R= 0.19, p<0.01), total N (R= 0.17, p<0.01), total C (R= 0.14, p<0.05), and exch. Ca (R= 0.21, p<0.01) correlated to differences in fungal community composition (Figure 5).

**3.5 Microbial taxonomic responses to soil depth**

Only 9% of bacterial ASVs and 2% of all fungal ASVs were consistently present in all soil depth samples (Figure A11). Upper
soil layers (0 to 20 cm) had the highest number of unique bacterial (20%) and fungal (32%) ASVs, suggesting that these surface layers supported a greater range of unique bacterial and fungal taxa than the lower depths. Less than 3% of bacterial ASVs and 1% of fungal ASVs were found to be unique within depth increments from the 40 to 100 cm soil layer; suggesting that subsoils did not harbour vastly unique or distinctive members of the microbial communities than those present in topsoil layers. Archaeal taxa belonging to Thermoplasmata, Desulfurococcales, Thermoprotei, Woesearchaeota *incertae sedis* (AR15, AR18,
AR16, AR20), Euryarchaeota, Candidatus Aenigmarchaeum, and Thermoproteales exhibited significant positive log-fold changes with depth (Figure 6). Furthermore, bacterial taxa belonging to Acidobacteria (e.g. Bryobacter, Gp20, Gp18, Gp25, Gp19, Gp6, Acidicapsa), Elusimicrobia (Blastocatella, Candidatus Endomicrobium), Planctomycetes (Phycisphaerales, Tepidisphaerales, Phycisphaerae), Firmicutes (Lactobacillales, Clostridia, Thermoanaerobacterales), Betaproteobacteria (Neisseriales, Ferritrophicales, Hydrogenophilales, Methylophilales) and Armatimonadetes (Gp2, Fimbriimonadales)
exhibited a positive log-fold change in subsoils (Figure 6). Fungal taxa which exhibited a positive log-fold change in subsoils (Figure 6) included Calcarisporiellomycota, Exobasidiomycetes (Entylomatales), Sordariomycetes (Microascales, Trichosphaeriales, Annulatascales, Ophiostomatales), Agaricomycetes GS29, Sareomycetes (Sareales), Orbiliomycetes (Orbiliales), Malasseziomycetes (Malasseziales), and Kickxellomycetes (Kickxellales and GS29).

In contrast, bacterial taxa which had a significantly negative log-fold change in subsoils (Figure 6) included Chlamydiae
(Chlamydiales), Bacteroidetes (Sphingobacteriales, Cytophagales), Acidobacteria (Edaphobacter, Candidatus Solibacter, Candidatus Koribacter, Gp5, Gp16, Gp1, Terriglobus), Gammaproteobacteria (Chromatiales), Firmicutes (Clostridiales, Bacillales), Gemmatimonadetes (Gemmatimonadales), Verrucomicrobia (Subdivision 3, Spartobacteria), Chloroflexi, Latescibacteria, and Actinobacteria (Ktedonobacterales, Actinomycetales, Gaiellales). The fungal phyla Mucoromycota, Rozellomycota, Mortierellomycota, and Glomeromycota exhibited large negative log-fold change values in subsoils, with
members of the classes Tremellomycetes (e.g. Tremellales, Filobasidiales, Trichosporonales), Umbelopsidomycetes (Umbelopsidales), Agaricomycetes (Thelephorales, Boletales), Leotiomycetes (Helotiales, Thelebolales), Sordariomycetes (Xylariales, Chaetosphaeriales, Sordariales) and Microbotryomycetes exhibiting particularly strong negative responses to increasing soil depth (Figure 6). More detailed outputs of the ANCOM-BC analysis are presented in Tables A8 to A13.





When correlating the abundance of microbial taxa to soil chemical properties, the microbial phyla Bacteriodetes, Chlamydiae,
Chloroflexi, and Mucoromycota were strongly correlated (Spearman rho > 0.7, p<0.05) with $^{14}$C abundance (Table A.15).
More specifically, strong positive correlations to $^{14}$C abundance were observed by the orders Candidatus Solibacter,
Subdivision3 unknown, Thelephorales, and Tremalles. In addition, the abundances of Tremellales were positively correlated
(Spearman rho > 0.7, p<0.05) with the total C & N content of the soil samples.

### 3.6 Structural changes in the soil microbiome with depth

The average node degree, clustering coefficient, and neighbourhood connectivity of interkingdom microbial networks were
significantly stronger in topsoil versus subsoil communities. However, the average path length within networks was
significantly higher in subsoil interkingdom networks (Table A15). For both topsoil and subsoil interkingdom networks,
bacterial genera were overwhelmingly the dominant community members compared with fungi and archaea. Compared to
subsoils, the topsoil interkingdom network had a more tightly clustered structure with a greater degree of positive co-
occurrences between genera (Figure 7). Furthermore, subsoil interkingdom communities exhibited a greater loss of co-
occurring fungal genera versus bacterial, highlighting the reduced dominance of fungi in the subsoil microbial community.
The network of fungal topsoil communities had a significantly (p<0.05) higher node degree, clustering coefficient,
neighbourhood connectivity, average shortest path length, and betweenness centrality than subsoil communities (Table A15).
In contrast, fungal subsoil communities exhibited a sparse network structure (Figure 7), with few positive co-occurrences
identified between genera. This was not apparent in the network structure of bacterial communities, where there were few
differences between topsoil and subsoils. Of all network characteristics, only average path length was found to significantly
differ (p<0.05; Table A15). Although there were taxonomic differences in the genera which had the highest node degrees
(Table A16), the main structural properties of topsoil and subsoil bacterial networks were similar.

### 4.0 Discussion

**4.1 Changes in soil C quantity, stabilisation, and age with depth**

Previous research has identified large stores of unaccounted C stocks located in subsoils (Balesdent et al., 2018; Gonzalez et
al., 2018; Ross et al., 2020). Our research identified that within Puruki Forest, 35% of total SOC stocks were allocated to soils
deeper than 30 cm. However, within just this 18 m sampling transect, C storage patterns were highly variable and ranged from
19 to 46% of total SOC stocks. Previously, Oliver et al. (2004) remarked upon the large variability in SOC storage patterns
across Puruki Forest, reporting average C stocks of 143 Mg ha$^{-1}$ to 1 m soil depth; values which are higher than observed in
our study (99.71 Mg ha$^{-1}$). High variability in subsoil C storage within Puruki Forest may be a signature of the natural textural
and structural variation observed in the area's pumice soils (Silver et al., 2000; Telles et al., 2003). Furthermore, this variability
may be an artifact of Puruki's previous land use history (native forest > pasture > plantation) and the harvesting disturbances
associated with historical management practices (Beets et al., 2002; Garrett et al., 2021; Rijkse and Bell, 1974). High variability





in SOC stocks across a relatively short spatial distance reaffirms previous research that attributes soil-carbon variability to much of the uncertainty in our ability to quantify subsoil C stocks (Ross et al., 2020). It is important to consider that this study was focused on a small-spatial scale, with soil samples obtained from 10 soil cores extracted along one transect within Puruki Forest. This variation needs to be considered for future studies.

Aligning with previous research findings (Garten, 2011; Wang et al., 2018), we observed an overall enrichment of soil $\delta^{13}$C

with depth. However, this was not a clear/linear enrichment; a highly variable $\delta^{13}$C soil layer was present at 70 to 90 cm depth. This finding was reflective of the high degree of structural variability in our soil cores, with layers of coarse pumice material (Taupo lapilli) present throughout the subsoil horizons below 50 cm (Froggatt, 1981; Rijkse and Bell, 1974). Wynn et al. (2005) identified soil texture as an important factor governing soil $\delta^{13}$C enrichment, with finer textured soils being associated with a greater degree of $\delta^{13}$C enrichment with soil depth than coarse-textured soils. Given Puruki Forest's previous land use

history from native forest, to pasture, and then planted forest (Beets and Brownlie, 1987), disturbance may have had an important role in altering soil $\delta^{13}$C with depth (Oliver et al., 2004).

Down to 1 m soil depth, $^{14}$C abundance declined by 229‰. While this is lower than the median global decline of 502‰ (Mathieu et al., 2015), the mean radiocarbon age of our deepest soil layers (~1570 yBP) corresponds to the age of the soil parent materials dated as 1760 ± 80 yBP before 1950 (Rijkse and Bell, 1974). The relatively young age of these is a result of

formation during the Taupo volcanic eruption. This can be directly observed by layers of Taupo lapilli on Waimihia, Rotoma, Waiohau, and Rotorua ash beds present in several of our subsoils.

## 4.2 Soil microbial diversity and abundance changed with soil depth

Subsoils held 49% of bacterial biomass and 33% of fungal biomass which corresponded neatly to values for subsoil C stocks. Soil microorganisms are responsible for the mineralisation of SOC and are influential in determining the formation,

composition, and persistence of SOC (Domeignoz-Horta et al., 2021; Jansson and Hofmockel, 2020). Although subsoil C is considered more resistant to microbial decomposition (Schmidt et al., 2011), an accumulation of microbial biomass in proximity to subsoil C stores presents concerns; particularly given our uncertainty on the effects of climate change on soil microbial activity and C substrate use (Yang et al., 2021). However, the contribution of microbial necromass C to total SOC is known to increase with soil depth (Ni et al., 2020). Thus, much of this accumulated microbial biomass in the subsoil may

be non-viable microbial cells. Our research does not provide information on the viability and activity of subsoil microbial biomass. Much like incorporating measures of SOC age and stability when quantifying subsoil C stocks, further research should measure the abundance of subsoil microbial cells which are viable and active, as well as those which are dormant but have the potential to become active under global change. Doing so will help us better investigate the vulnerability of subsoil C stocks to microbial decomposition under climate change.

In agreement with the findings of previous research (Blume et al., 2002; Eilers et al., 2012; Mundra et al., 2021; Rosling et al., 2003), soil microbial diversity and abundance declined sharply with depth. These declines occurred alongside clear shifts in microbial community composition. Below 30 cm, the magnitude of these changes with depth slowed which marked a clear





transitional change between the topsoil-subsoil boundary. Although fungal diversity and biomass (inferred via rRNA gene abundances) were generally much lower than that of bacteria, patterns in response to depth were consistent across both
microbial kingdoms. The changes in microbial diversity and abundance were highly correlated with the quantity and age of soil C and the C:N ratio. These findings support research that has proposed declines in soil C quantity, quality, and availability to be strong drivers of declines in microbial diversity and abundance (Eilers et al., 2012). Due to their proximity to aboveground vegetation, topsoil layers typically receive a wider load and range of fresh C inputs from surface litter and plant roots (Fierer et al., 2003; Spohn et al., 2016). This may explain why soil depths of 0 to 20 cm supported the greatest number of unique
ASVs (i.e. those which occurred only in that soil layer), with the number of unique ASVs declining sharply with depth.

**4.3 Shifts in community co-occurrence patterns with depth**

Whilst bacterial diversity and abundance declined sharply with depth, bacterial subsoil communities exhibited only minor differences in network structure compared to topsoils. Subsoil bacterial networks had significantly higher average shortest path lengths, which suggests that positively co-occurring genera were more exclusively coupled together in subsoils versus
topsoils (Xu et al., 2021). A more exclusive coupling of subsoil bacteria may be a result of environmental filtering imposed by the subsoil environment, as only soil bacteria tolerant to the resource-limited environmental conditions can populate this habitat. Compared to bacterial communities, subsoil fungal networks exhibited greater structural differences between topsoil and subsoils. Subsoil fungal networks were almost entirely disconnected and there was no suite of co-occurring fungal genera that characterised the subsoil microbiome. The breakdown in subsoil fungal network structure may be explained simplistically
by reduced community size limiting the ability for fungal co-occurrences to establish. Microbial density declines with soil depth (Spohn et al., 2016) and only 31 core fungal genera were present in the subsoil networks. In contrast, 178 bacterial genera were present in subsoil networks meaning their relatively larger community abundance may have contributed to the lesser effects of soil depth on their co-occurrence patterns. Furthermore, as the properties of soil fungal communities are strongly driven by characteristics of aboveground vegetation (Likulunga et al., 2021; Urbanová et al., 2015), the breakdown
in subsoil fungal co-occurrences may be due to their increasing physical distance from the surface litter and plant rhizosphere.

**4.4 Differential responses of microbial taxa to topsoil vs subsoils**

Consistent with previous research, subsoils had an increased abundance of numerous archaeal taxa from Euryarchaeota, Crenarchaeota and Woesearchaeota (Brewer et al., 2019; Eilers et al., 2012; Frey et al., 2021; Hartmann et al., 2009). Proposed reasons for the increased abundance of archaea with depth include their adaptation to chronic energy stress (Valentine, 2007);
their involvement as ammonia oxidizers in driving autotrophic nitrification in deep soil layers (Eilers et al., 2012); their preference as methanogens for anaerobic environments (Feng et al., 2019); and their adaptation to nutrient-poor environments as slow growing oligotrophs (Turner et al., 2017).

Also aligning with previous research, several representatives of the bacterial phyla Acidobacteria and Firmicutes increased in abundance with depth which may be explained by their tolerance for nutrient-limited environments (Bai et al., 2017; Brewer





et al., 2019; Frey et al., 2021; Feng et al., 2019; Lladó et al., 2017). However, several representatives of Acidobacteria and Firmicutes also declined with depth, such as Candidatus Solibacter, Candidatus Koribacter, and Clostridiales. Hansel et al. (2008) reported high variability in the distribution of Acidobacteria subdivisions amongst different soil horizons. The broad phylogenetic coverage and ecological diversity of Acidobacteria can make it difficult to infer metabolic and functional adaptations based on their taxonomic position alone (Hansel et al., 2008). Such issues highlight the need to adopt more trait-

based approaches when investigating the microbial taxa that exhibit adaptive responses to soil depth. For example, the large genome size of Candidatus Solibacter (Challacombe et al., 2011) may explain its high abundance in surface soils, which have a wide range of C and nutrient inputs and greater exposure to environmental flux (Barberán et al., 2014). Theoretically, soil depth may drive a reduction in microorganisms with large genome sizes, as specialisation becomes more favourable under harsh conditions and maintenance of a larger genome becomes too energetically costly.

Strong declines in the abundance of Bacteroidetes with depth have been previously well reported (Eilers et al., 2012; Feng et al., 2019; Mundra et al., 2021) and attributed to Bacteroidetes' copiotrophic behaviour. In our study, the abundance of Bacteroidetes positively correlated with $^{14}$C abundance. This indicates an association with fast cycling, bioavailable C which typically declines with depth as more chemically resistant, slow cycling C forms a greater proportion of the C pool (Koarashi et al., 2012; Torn et al., 1997). The abundance of Chlamydiae also declined with depth and was positively correlated to younger

SOC. This phylum has been positively associated with lower soil pH, organic matter content, and C:N ratio (Zhalnina et al., 2015; Ma et al., 2021), properties that commonly change with depth. Thus, declines in Gemmatimonadetes and Chloroflexi with depth is inconsistent with previous research (Bai et al., 2017; Chu et al., 2016; Frey et al., 2021; Li et al., 2020; Seuradge et al., 2016; Zhang et al., 2019). In previous studies, the abundance of Gemmatimonadetes have increased with depth, and this has been attributed to preferences for low soil moisture conditions (Debruyn et al., 2011; Frey et al., 2021). However, the

highly permeable soils within Puruki Forest exhibit a greater potential for soil moisture to increase with depth, with the subsoil horizons having a high moisture capacity (Beets and Brownlie, 1987; Will and Stone, 1967; Beets and Beets, 2020). Thus, the decline of Gemmatimonadetes with soil depth at Puruki Forest may be due to its preference for low soil moisture conditions. Soil depth was associated with large declines in fungal taxa (i.e. Mucoromycota, Rozellomycota, Mortierellomycota, and Glomeromycota) previously reported in the literature as saprophytic and rhizospheric fungi (Bonfante and Venice, 2020;

Hoffmann et al., 2011; Wagner et al., 2013; Yurkov et al., 2016). Understandably, the abundance of saprophytic fungi would decline with increasing physical distance from plant litter inputs on surface soil layers. Carteron et al. (2021) observed large shifts in the ratio of saprophytic: mycorrhizal abundance with soil depth, which correlated to changes in soil chemistry. In our study, the abundances of Mucoromycota, Thelephorales, and Tremalles were correlated to soil C quantity and age.

**5.0 Conclusion**

At least 35% of total SOC stocks within Puruki Forest are present in subsoil layers below 30 cm, with this value reaching up to 49%. In the deepest soil layers, the mean C age was 1571 yBP, corresponding closely to the estimated age of soil parent



materials. Although there were large declines in microbial biomass and diversity with soil depth the magnitude of these changes slowed past 30 cm depth, marking a topsoil-subsoil transitional boundary in the soil profile. Despite the sharp vertical declines in microbial biomass, 49% of the bacterial and 33% of the fungal biomass was allocated to the subsoil. Quantifying the viable activity of this microbial biomass is essential for predicting the vulnerability of subsoil C stocks to microbial decomposition under climate change. Numerous archaeal taxa exhibited an increased abundance in subsoils, indicating their potential ecological importance in deep soil environments. Furthermore, numerous fungal and bacterial taxa exhibited significant differential abundances between topsoil and subsoil layers. This supports the notion that subsoil environments act as an environmental filter; with the vertical distribution of soil microorganisms reflecting the sharp changes in soil physical and chemical properties with depth. Given the size and age of the sub-surface carbon pools, the deep soil carbon contributes strongly to total forest carbon budgets but is currently unassessed in carbon models. Furthermore, the potential susceptibility of ancient forest-soil carbon to change in land use and climate will be important to integrate into earth system models. The results of this research contribute to our understanding of the biological and chemical properties of productive forest subsoil environments, a critical research gap that is fundamental for assessing the vulnerability of forest SOC stocks to climate change.

**Code/Data availability**

The R code used for performing the microbial diversity analyses is available upon request.

**Author contribution**

Alexa K. Byers: Methodology, Formal analysis, Data curation, Writing- Original Draft, Visualization. Loretta G. Garrett: Methodology, Writing- Review & Editing, Supervision, Project administration. Charlotte Armstrong: Methodology. Fiona Dean: Methodology, Resources. Steve A. Wakelin: Conceptualization, Writing - Review & Editing, Supervision, Project administration, Funding acquisition.

**Competing interests**

The authors declare that they have no conflict of interest.

**Acknowledgements**

Thank you to all the staff at Scion analytical laboratory services (Scion, NZ), Angela Wakelin (Scion, NZ), Stephen Pearce (Scion, NZ), Jeff Hatten (Oregon State University, US), and Katherine Heckman (U.S Department of Agriculture, US) for your invaluable contributions towards this research project.





**Funding**

Funding for this research came from New Zealand Ministry of Business, Innovation & Employment (MBIE) Strategic Science
Investment Fund held by Scion (C04X1703).



**Figures and Tables**

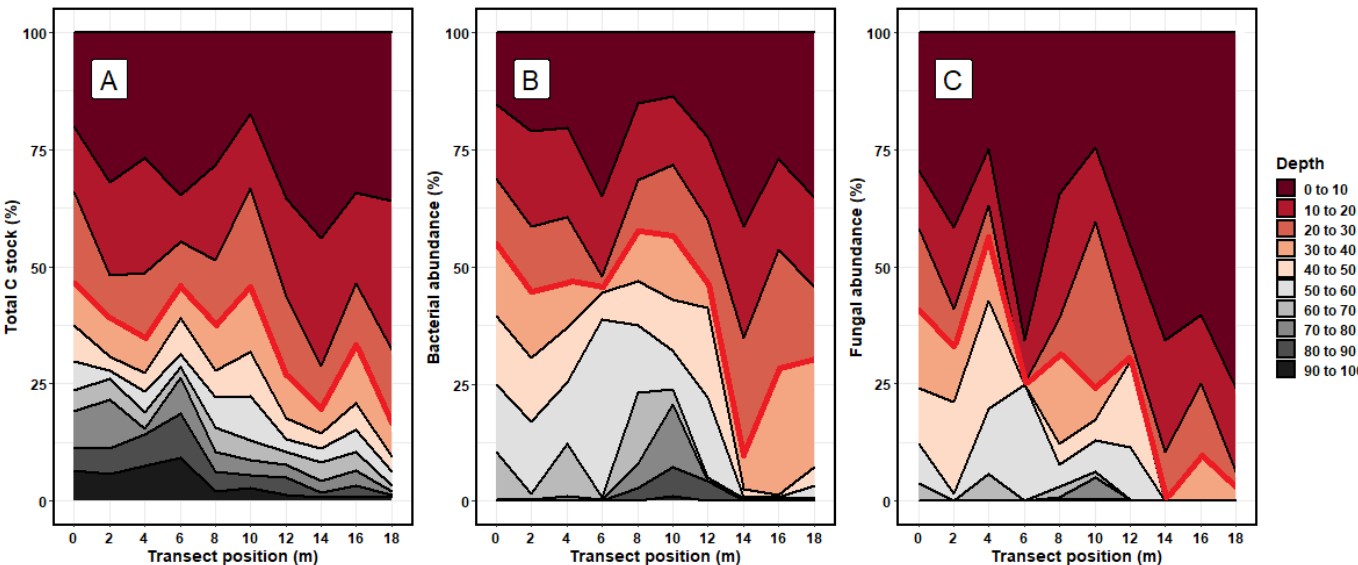

**Figure 1. The relative proportion (%) of (a) total carbon stocks (C Mg ha$^{-1}$), (b) bacterial biomass (16S rRNA copy number) and (c) fungal biomass (ITS rRNA copy number) allocated to each depth increment down the 1 m soil cores sampled from each transect. Red line shows typical 'cut off' point of topsoil subsoil boundary, where measures below 30 cm are rarely taken.**






**Figure 2. The mean ± SD changes in total C stocks (Mg ha$^{-1}$), δ$^{13}$C [‰], Δ$^{14}$C [%] and CRA [yBP] with soil depth.**



**Figure 3. The mean ± SD changes in bacterial and fungal chao1 richness, Shannon diversity, Pielou's evenness, and copy number/ng DNA with soil depth.**







**Figure 4. Pairwise Spearman's correlation showing the strength in correlation between different soil parameters and measures of microbial diversity and abundance. Blank cells indicate non-significant (p>0.05) correlation values.**





**Figure 5. NMDS ordination plots calculated using Bray Curtis dissimilarity matrices showing the differences in (A) bacterial and (B) fungal community composition between the different soil depth increments. C and D show envfit plots show soil parameters as fitted vectors which significantly correlated (p<0.05) to differences in (C) bacterial and (D) fungal community composition when tested using Mantel tests.**




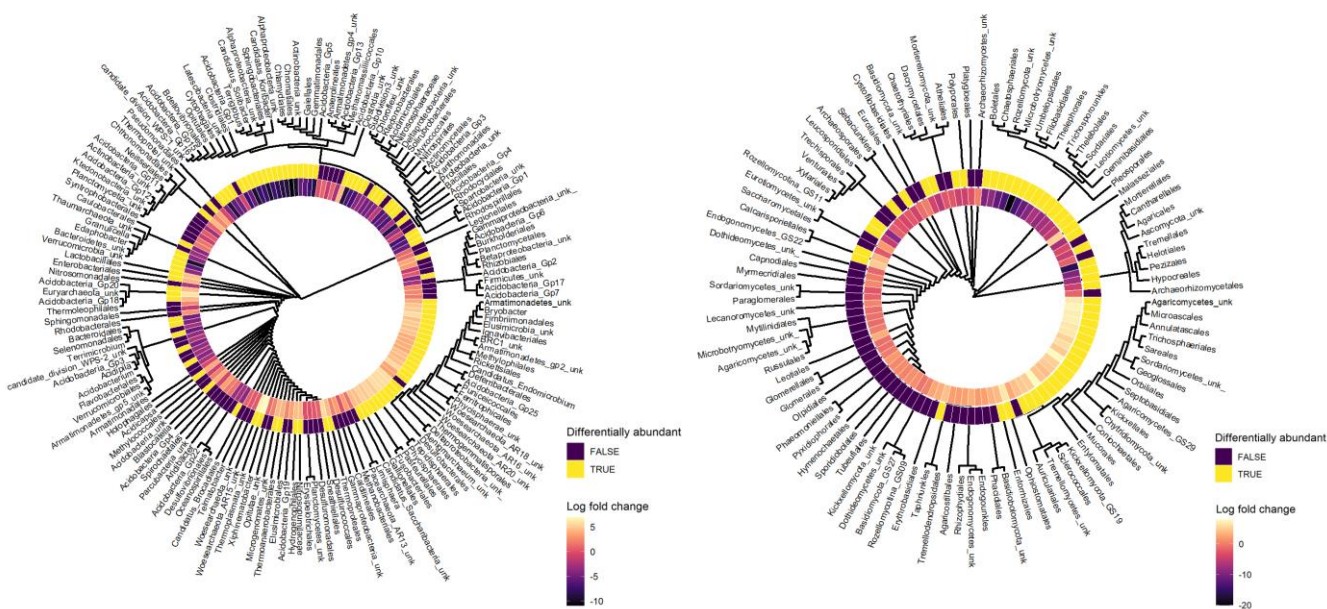

**Figure 6. Cluster dendrograms of bacterial and fungal orders based on their bias adjusted abundances (ANCOM-BC analysis). Hierarchical clustering was performed on Euclidean distances of bias adjusted abundances using the average linkage method. Each taxa's log fold change (LFC) value between topsoil and subsoil is displayed. Taxa with positive LFC values were higher in subsoils and taxa with negative LFC values were lower in subsoils. The statistical significance of their differential abundance between topsoil and subsoil is displayed, with TRUE = p<0.05 and FALSE = p>0.05.**





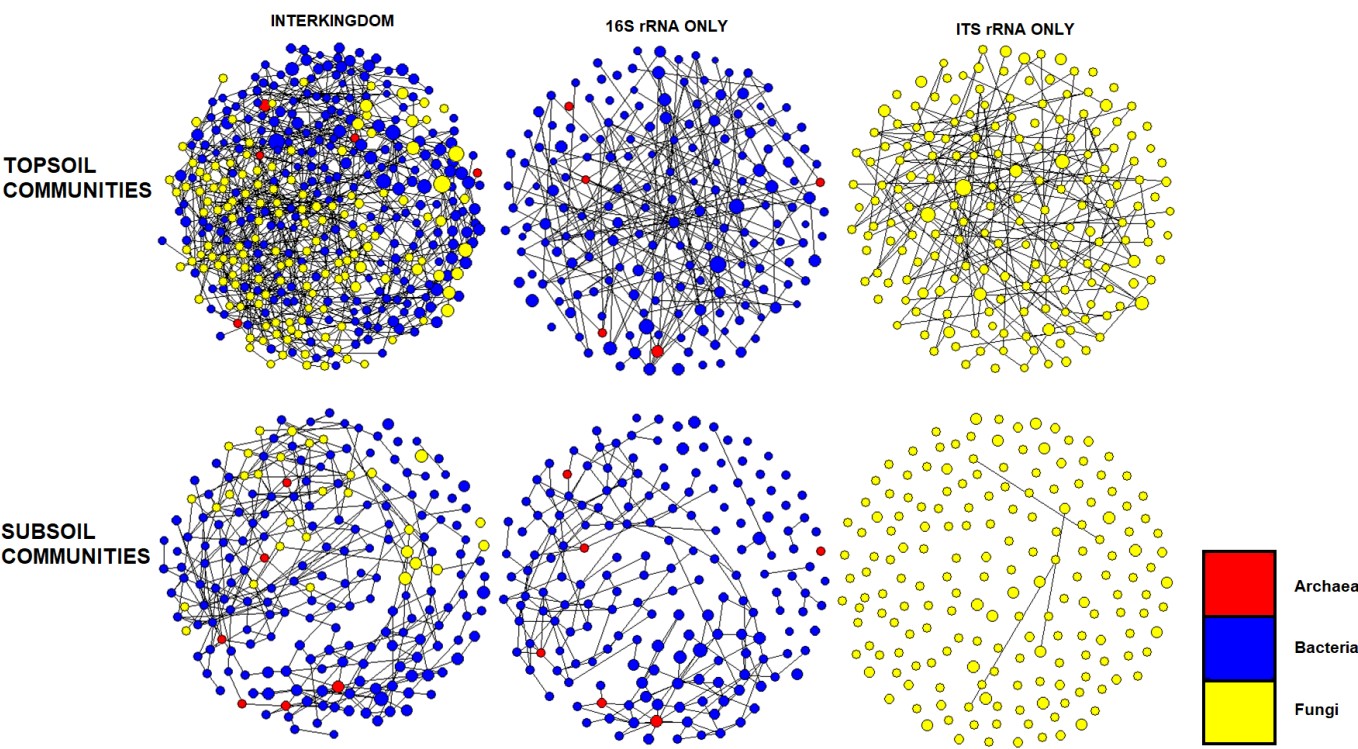

**Figure 7. SPIEC-EASI networks constructed on the core microbial genera associated with topsoil (0 to 30cm) and subsoil (30 to 100cm) layers. Negative edge weights were set to zero, thus edges between nodes represent only positive correlations between genera. Node size represents the normalized (centred log ratio) mean abundance of its respective genus**



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
