# Peer review of "Soil depth as a driver of microbial and carbon dynamics in a planted forest (*Pinus radiata* D. Don.) pumice soil"

_EGUsphere, 2022_

## Author Response (AR1)

**Author's response**

Thank you to the topical editor and two reviewers for contributing their time and expertise to reviewing our manuscript- it is greatly appreciated. Our responses to the editor and reviewer's comments below outline the changes that have been made to our revised manuscript as a result of the review process.

**Responses to topical editor**

*"Please go ahead and revise the manuscript as outlined. Thank you for the detailed response to reviewer comments.*

*It is for the authors to decide whether to include negative interactions in the network analysis. I think reviewer 1 was suggesting to include the justification in the methods, regardless of your choice."*

- Response: We have updated the networks to include negative interactions in the revised manuscript. Originally, we only included positive edge weights (interactions) in the networks as we were interested in identifying positive species co-occurrences. However, upon revisiting the network analysis, including negative edge weights (which signal negative interactions) is valuable to the analysis, particularly when studying nutrient limited subsoil environments where biotic competition may be high.

*"Again, in terms of the chemical data and Data in Brief article it is an authors' decision to make. For this paper, if you mention that total soil P was measured, please provide the results."*

- Response: The data for total P is now available in the revised manuscript and supplementary materials. This includes statistical analyses to measure changes in total P with depth, as well as incorporating total P into the microbial community analysis.

*"I agree with reviewer 2 that the last two lines of the abstract are too wide to describe the implications of the results. Can you make them more specific such that you project a more targeted outlook of this study?"*

- Response: The final sentence of the abstract has been modified to be more specific to the findings of our study- "These research findings highlight the importance of quantifying subsoil C stocks for accurate C accounting. By performing a broad range of analytical measures, this research comprehensively characterised the abiotic and biotic properties of a subsoil environment - a frequently understudied but significant component of forest ecosystems." Lines 23 to 26.

*"I read the following line 4 times and it still doesn't make sense to me, including the phare 'having a physical and chemical nature'. "When undisturbed, deep SOC stocks are considered to represent the 'stable' fraction of the SOC pool, having a physical and chemical nature that is, while in situ conditions remain constant, resistant to decomposition by soil microorganisms." Please consider the reviewer comments to improve readability."*

- Response: This sentence has been edited to "Deep SOC is more resistant to decomposition by soil microorganisms due to the biological, physical, and chemical conditions of the subsoil environment (Schmidt et al., 2011). Thus, when undisturbed,

deep SOC is considered to represent the 'stable' fraction of the SOC pool." We hope this has improve the readability of this section. Lines 36 to 38.

*"Please ignore the science fiction comment made by reviewer 2."*

*The title is slightly misleading: 'microbial communities are present deep in soils' is not a finding worthy of highlighting. Please consider alternatives."*

- Response: The title of the manuscript has been changed to "Soil depth as a driver of microbial and carbon dynamics in a planted forest (*Pinus radiata* D. Don.) pumice soil".

**Responses to reviewer's comments**

*Note: Detailed responses to several of these comments were provided in the previous author response submission.*

Reviewer 1
*General comments*

*The authors examine an important topic: what is the size and age and variability of soil organic carbon (SOC) in deeper portions of the soil profile (30 -100 cm in depth e.g. subsoil), and how do these quantities vary across space? The authors description of SOC and soil microbe composition changes along the soil profile will inform ongoing scientific discussions of soils and their ability to store carbon (C). This question and the findings of this study are also relevant for the growing number of projects (public and private) hoping to mitigate climate change through increasing SOC pools via soil amendments or ecological restoration.*

*The methods used by the authors are appropriate for investigating this topic. I am also pleased by their use of multiple estimates of SOC age - which gives confidence to their findings. I found the manuscript to be well written, maintaining a healthy balance of thoroughness and interest throughout the text.*

- Response: See response provided in a previous author response submission.

*Specific comments*

*To understand and generalize from these dynamics, I (and I assume readers) would be interested to know the average and range of pH in the soils of this planted forest. I would add this information to the description of this site (which is otherwise fairly comprehensive).*

- Response: The requested information has been added to the revised manuscript. Lines 81 to 82.

*The authors need to pay attention to the notation that they use and keep this consistent throughout the text and figures. At times they switch between percent and permille for 14C (Fig. 2). Figure axis titles should also be the appropriate symbol, and not %o for permille.*

- Response: These errors have been corrected in the revised manuscript. See Figure 2.

*CRA is not defined in the text. Please do so before introducing this as a measurement.*

- Response: Conventional Radiocarbon Age (CRA) has now been defined properly in the revised manuscript. Lines 128 to 129.

*My understanding of Fig. 5 is that C and D are replots of A and B, but just now with the environmental vectors overlaid. These replots with the environmental vectors don't allow me to interpret the shifts in your data points (the data is too scrunched near 0,0). I suggest either removing C and D or re-scaling the vectors (divided by 10 maybe) and replot A and B with these rescaled vectors so that readers can see how these environmental variables are affecting your estimated microbial compositions.*

- Response: Subplots C and D have been removed from Figure 5 in the revised manuscript. See Figure 5.

*Fig. 6 has some bizarre mispellings of names (Sebciunkles instead of Sebacinales) and I would standardize the names to remove the trailing '_ unk' artifacts of taxonomic clustering*

- Response: Errors have been corrected and edits to names made in the revised manuscript. See Figure 6.

*Line 175 - McMurdie not Mcmurdie*

- Response: Correction made. Line 176.

*It's not clear to me why you exclude negative interactions from the network analysis. If you can justify this briefly, do so in the text.*

- Response: See response to editor comment.

*Line 414/415- you didn't examine microbial biomass though, you quantified DNA and you've stated that you did not partition between viable and nonviable cells/hyphae. While DNA abundances can be used as a proxy for biomass in controlled systems of relatively short age I don't agree with claiming this as biomass here, as much of the DNA you sampled from these lower depths may actually be relictual. Best refer to it as something neutral like 'DNA abundance'*

- Response: Revisions have been made based on the reviewer's suggestion (edits throughout entire manuscript).

*I found it interesting that Bray P did not correlate with other factors (Fig. 4). In the text you mention that you also measured total soil P, however it looks like this wasn't included in analyses. Was this estimate just not variable? I noticed that the soils are very young (from a recent volcanic eruption even), so I'm assuming that the microbes and vegetation are more N limited while P is abundantly available? If it's not too much trouble I'd add general P and N abundance or availability at this site in the site description (plant-microbe people love this). Total P should atleast show up in the supplemental materials along with total C and N(e.g. Table A2)*

- Response: Revisions made to include total P data. See response to editor comment.

Reviewer 2

*This study explored the changes in SOC dynamics and soil microbes in the Puluki Experimental Forest (New Zealand) down to a soil depth of 1 m. ITS and 16S rRNA*

*sequencing and quantitative real-time PCR were used to measure changes in soil microbial diversity, composition, and abundance. Stable (δ13C) and radioactive (14C) C analyses were performed to assess depth-driven changes in SOC stability and age. It has to be said that it is very important to use these methods to explore the dynamic changes of deep SOC, but it is not the most innovative. The sentences are smooth and comfortable to read. To my surprise, the total C stocks of deep soil accounted for only 35%, which was not more than 50% as previously reported. The Introduction is like science fiction, lacking clear questions and hypotheses. Moreover, few links were made between changes in soil microbes and soil carbon stocks in the Discussion. Other suggestions are as follows:*

- Response: See response provided in a previous author response submission.

*Title: Microbial communities are not suitable to be described as big and old.*

- Response: Title of the manuscript has changed. See response to editor comment.

*L16: Why must it be an incremental change?*

- Response: See response provided in a previous author response submission.

*L22: Does soil carbon refer to carbon storage or carbon concentration or stability or others?*

- Response: The sentence in question is no longer in the abstract but has been updated to "Our research identified large declines in microbial diversity and abundance with soil depth, alongside significant structural shifts in community membership". Lines 19 to 21.

*L23-L25: This study is only a sample study, why do you say "These research findings highlight the importance of quantifying subsoil C stocks for accurate systems-level global and local C budgets and modeling"? Moreover, this study does not address climate change.*

- Response: See response to editor's comment.

*L28-31: These three sentences all emphasize the importance of forest soil carbon, which can be simplified and combined with the next paragraph.*

- Response: Revision made based off reviewer suggestion. Lines 28 to 30.

*L41-42: What is the meaning of "having a physical and chemical nature"?*

- Response: Revision made, see response to editor's comment. Lines 36 and 38.

*L44-46: Although these are all factors affecting the stability of SOC, they are not addressed in this study, and the preamble should introduce more advances in microbes, isotopes, etc.*

- Response: See response provided in a previous author response submission.

*L46: delete ", thereby,"*

- Response: Deletion made.

*L51: delete "fundamental"*

- Response: Deletion made.

*L64: How to understand the meaning of the word " fundamental"?*

- Response: Term removed from revised manuscript.

*L65: How to understand "at a highly refined spatial scale"?*

- Response: Revision made to reduce ambiguity. Lines 59 to 60.

*L87: delete the second "(".*

- Response: Deletion made.

*L100: To avoid DNA degradation, soil samples are usually stored at -20 or -80 â before DNA determination. How long will this study complete DNA determination after sampling?*

- Response: See response provided in a previous author response submission.

*L103: What are the "Mehlich 3 extractable elements"? How are fractions equal to 2 mm treated?*

- Response: See response provided in a previous author response submission.

*L150: How to calculate soil carbon storage, it is recommended to list a formula.*

- Response: Formula for calculation of soil carbon stocks now included in revised manuscript. Lines XX to XX.

*L153: What are the "Bray P sequential 1"? What does slope corrected mean?*

- Response: See response provided in a previous author response submission.

*L209: It is recommended to merge 3.1 and 3.2.*

- Response: Sections have been merged in revised manuscript.

*L213: Is the result reliable if the variation of SOC stocks in the coarse soil fraction is so large? Some similar results can be described together to avoid redundancy. For example, the results of most indicators decrease with the soil layer. Where is the Table?*

- Response: See response provided in a previous author response submission.

*L220: delete "this trend was variable. In particular,".*

- Response: Deletion made.

*L225: It is suggested to delete some results in 3.4 and 3.5. Not all results need to be described, but they should be targeted.*

- Response: Revisions made throughout sections 3.4 and 3.5.

*L251-257: Generally, |R|<0.3 is considered a weak correlation, where R is generally lower than 0.3. I doubt the reliability of the results.*

- Response: See response provided in a previous author response submission.

*L264: Microbial classification names need to be italicized.*

- Response: See response provided in a previous author response submission.

*L333: neatly to? Or nearly to.*

- Response: Word removed from revised manuscript.

*L350: What does "quantity" mean here?*

- Response: See response provided in a previous author response submission. Revisions made in revised manuscript. Line 345.

*L365: How to calculate the microbial density?*

- Response: See response provided in a previous author response submission.

*L367-368: The abundance of bacteria significantly declined with soil depth. Is it contradictory to say that the relatively large community abundance?*

- Response: See response provided in a previous author response submission. Revisions made in revised manuscript. Line 357.

*L408: How to understand that the abundances of Myxomycetes, Thelephorales, and Tremells were related to soil C quantity and age?*

- Response: See response provided in a previous author response submission.

*The red line in Figure 1 is not obvious, so it is recommended to consider other colors.*

- Response: Red line has been changed to yellow line in Figure 1.

---

## Author Response (AR2)

Dear Editor,

Thank you for taking the time to review our manuscript. Your feedback is greatly appreciated. We have revised our manuscript in agreement with your comments and the responses to these comments can be seen below. We hope that these revisions satisfy the journal's requirements and look forward to hearing about the outcome of our manuscript submission.

Kind regards,

Alexa-Kate Byers on behalf of the authorship team.

**"Comments to the author**:
I am happy with the revisions done by the authors. I have some minor comments.
Line 62, 125: 16S/ITS rRNA should be 16S rRNA/ITS. ITS is the spacer DNA situated between the small-subunit rRNA and large-subunit rRNA genes.

- **Response:** All instances where 'ITS rRNA' was used have been corrected to state just 'ITS'.

I would suggest excluding the scientific name - (Pinus radiata D. Don.) - in the title but this is up to authors to decide.

- **Response:** 'D. Don' has been removed from the title in the revised manuscript.

Line 310: 'significantly ___ than that'; missing word

- **Response:** Apologies, we have inserted the missing word and the sentence now reads "The average node degree and neighbourhood connectivity of the topsoil interkingdom networks was significantly **higher** than that of the subsoil networks." Line 278 to 279.

Lot of repeated lines eg. line 43, line 306, line 390. Is this a formatting error?

- **Response:** these sections of the revised manuscript have been made more concise and avoid the use of repetitive language.

Please go through the entire manuscript carefully to spot grammatical and formatting errors. I did not read meticulously yet was able to spot some errors/inconsistencies."

- **Response:** Apologies for the errors, we have gone through the entire manuscript and made grammatical corrections where required.